# The last stretch: Barriers to and facilitators of full immunization among children in Nepal's Makwanpur District, results from a qualitative study

**Alicia M. Paul** [1]*, **Shraddha Nepal**[2], **Kamana Upreti**[2], **Jeevan Lohani**[2], **Rajiv N. Rimal**[1]

**1** Department of Health, Behavior and Society, Johns Hopkins University Bloomberg School of Public Health, Baltimore, Maryland, United States of America, **2** Nepal Evaluation and Assessment Team, Kathmandu, Bagmati Province, Nepal

* apaul17@jhu.edu

**Data Availability Statement:** All interview and focus group transcript files are available from OSF Registries. DOI: 10.17605/OSF.IO/GKX3F.

## Abstract

### Background

Approximately 35% of Nepal's children have not received all recommended vaccines, and barriers to immunization exist on both the demand- (i.e., access, affordability, acceptance) and supply- (i.e., logistics, infrastructure) sides.

### Objective

This article describes a formative study to understand the barriers to and facilitators of immunization in Makwanpur, Nepal from both the demand- and supply-sides.

### Methods

Through in-depth interviews, key informant interviews, and focus group discussions ($N = 76$), we assessed knowledge, attitudes, and experiences with immunization; social norms related to immunization; perceptions of local health facilities; and descriptions of client-provider relationships. Data were analyzed using an iterative, grounded theory approach.

### Results

Three major themes emerged, including positive demand of vaccines, lack of mutual trust between service seekers and service providers, and internal and external motivators of vaccine supply. On the demand-side, caregivers reported high levels of immunization-related awareness, knowledge, and acceptance, largely perceived to be due to a generational shift. On the supply-side, providers expressed passion for their work despite lack of support from local authorities and a desire for more training. Between caregivers and providers, lack of mutual trust emerged as a prominent barrier, revealing a cycle of positive service bias.

**Funding:** This work was supported by The Bill and Melinda Gates Foundation [ID number OPP1212270], awarded to RR. The funders had no role in study design, data collection and analysis, decision to publish, or preparation of the manuscript. https://www.gatesfoundation.org/.

**Competing interests:** The authors have declared that no competing interests exist.

## Conclusions

We identified mutual trust as a key pathway toward reaching full immunization coverage in Nepal and we recommend future interventions adopt an approach which focuses on removing social barriers (i.e., distrust) and structural barriers (i.e., opening hours, neglected infrastructure) to immunization.

## Introduction

Vaccination reduces childhood mortality, saving two to three million lives each year [1]. Despite global increases in childhood vaccine uptake, rates remain sub-optimal in many parts of the world, and vaccine-preventable diseases continue to pose a significant public health threat [1]. Currently, 20 million children worldwide have not received the minimum basic vaccines, and only 11% have received the full schedule recommended by the World Health Organization [2].

In Nepal, approximately 65% of children between 1–2 years of age have received all recommended vaccines [3]. However, only 43% have received their vaccines at the age-appropriate times for which they are scheduled [4]. Moreover, significant disparities exist regarding who receives vaccines. For instance, the gap in coverage between those whose mothers' have attained greater than secondary education and those whose mothers have received no education is 23%, and the corresponding gap between those in the Brahmin/Chhetri castes, which are considered more privileged castes, and those in the Terai/Madhesi castes is also 23% [5]. Although infant mortality in Nepal has substantially declined from 78 to 32 deaths per 1,000 in the last two decades, more work is needed to further reduce child mortality, and addressing deaths due to vaccine-preventable diseases could be a key pathway toward achieving that progress [4].

Nepal initiated the National Immunization Programme (NIP) in 1977, originally offering Bacillus Calmette–Guérin (BCG) and Diphtheria, Pertussis, and Tetanus (DPT) vaccines to children free of charge [6]. Today, NIP includes vaccines for 11 vaccine-preventable diseases–diphtheria, hemophilus influenza B, hepatitis B, Japanese encephalitis, measles, pertussis, pneumococcal disease, polio, rubella, tetanus, and tuberculosis [7]. NIP is a top priority program in Nepal and has achieved significant milestones, including being declared polio-free in 2014 and maintaining maternal and neonatal tetanus elimination status since 2005 [6, 7]. In 2012, Nepal began the "Reaching Every Child" program under NIP to declare full immunization across the country. By 2020, 56 out of 77 districts had achieved this goal. However, 26 districts including Kathmandu, the capital of Nepal, have immunization coverage rates of 80% or less and 14 districts have dropout rates greater than 10% [6]. Nepal still has a long way to go to achieve its goal of 95% immunization coverage by 2030 [6].

### Immunization barriers and facilitators

Barriers to and facilitators of immunization exist on both the supply-side (which includes logistics, such as cold chain management) and the demand-side (which includes access to and affordability of vaccines and psychosocial factors like fear and acceptance) [8]. On the demand-side, several health service-related behaviors of parents and caregivers are associated with greater immunization uptake, including retaining the child's immunization card [9–11], obtaining services at privately-owned facilities [10], and giving birth in an institutional setting [9, 12]. Institutional delivery is particularly correlated with immunization coverage, as health

workers can more readily administer the first scheduled vaccine (BCG) immediately after birth than in a home setting [9, 12]. Additionally, giving birth in a medical facility may offer parents an additional opportunity to learn about the benefits of vaccines.

Awareness of the vaccine schedule is a crucial aspect of parent knowledge [10, 11, 13]. Nepal requires seven separate visits over 15 months to complete the full schedule [12]. As such, parents who are less familiar with this schedule are at risk for forgetting or delaying vaccine appointments [9].

Immunization may be hindered by several factors, including parental fear and health system barriers. Several studies suggest that parental fear could be contributing to low immunization rates by creating poor perceptions of vaccines [11–15]. Both in the US and globally, parents have reported considerable distrust in vaccines, which has created a pattern of vaccine hesitancy [16, 17]. Some studies have found that parents may view their children as being "too ill" to receive a vaccine [12, 14] or fear that the side effects will outweigh potential benefits [11, 13, 15]. Indeed, some vaccine-preventable diseases have re-emerged in both developed and developing countries due to parental fear.

On the health system side, factors such as procurement and supply of vaccines, as well as provider competency and communication, have been found to influence vaccine coverage. For example, Rainey et al. [18] revealed in their systematic review that the most common reason associated with under-vaccination is the immunization system. More specifically, issues of inadequate access to services, long distance to immunization centers, poor health worker knowledge and attendance, and vaccine availability are strongly linked with under-vaccination in low- and middle-income countries [18]. Favin et al. [15] also reported a concerning pattern–that mothers are often treated in a disrespectful and, in some cases, abusive manner by health workers at immunization appointments–which discouraged them from future visits and left them with unanswered questions about their children's health.

## Study objectives

This study presents findings from the formative assessment of the Rejoice Architecture Project (see Paul et al. [19] for the study protocol), a social norms-based intervention aiming to improve child vaccination in the Makwanpur District in Nepal. To do so, the project addresses the physical, structural, and social barriers within health facilities at the nexus of both demand- and supply-sides; namely, between maternal caregivers and health workers. In this study, we envisioned the healthcare clinic as the site that straddles both the demand- and supply-side issues that could result in reduced visits and, by extension, reductions in immunizations. In particular, we sought to understand the demand- and supply-side barriers to and facilitators of immunization in Makwanpur District, Nepal.

## Materials and methods

This study adopted a grounded theory design using in-depth interviews, key informant interviews, and focus group discussions. This study was approved by the Institutional Review Boards with Johns Hopkins University in the United States (no. 9951) and the Nepal Health Research Council (no. 860/2019).

### Study setting

Administratively, Nepal is divided into seven provinces, each of which is further divided into districts (77 in total), then municipalities (called a "Palika" in the local vernacular, which is also the term we use in this paper) and, finally, wards. This study was conducted in three Palikas (Thaha, Kailash, and Bakaiya) in the Makwanpur District. Makwanpur has a population of

approximately 460,000, of which 4% are between 0 to 23 months of age and 22% are married women of reproductive age [20]. Makwanpur is considered a Category 1 District in terms of immunization access (DPT-HepB-Hib1 coverage) and utilization (DPT-HepB-Hib1 vs DP-HepB-Hib3 dropout), meaning it has approximately 80% coverage and less than 10% dropout [21]. As of 2020, the percentage of children fully immunized as per NIP schedule in Makwanpur was 68%, comparable to the national rate of 65% [3].

Makwanpur was selected for this study for its representation of Nepal's ethnic and geographic diversity and large size. For example, Makwanpur represents 78 of Nepal's ethnic groups, inclusive of various cultures and languages [22]. In addition, the three selected Palikas were chosen to represent Nepal's three regions of the mountains (Thaha Municipality), hills (Bakaiya Rural Municipality) and terai (lowlands; Kailash Rural Municipality). Lastly, Makwanpur has a land area of approximately 2,426km², about 1.6% of Nepal's total land area [23].

## Participants

Seventy-six individuals participated in this study across the catchment areas (1-kilometer radius) of five health facilities. All participants were 18 years or older and were either permanent residents of the health facility's catchment area or worked for, oversaw, or volunteered at the health facility. Ten in-depth interviews were conducted with five fathers and five grandmothers of children under the age of 2 (see Table 1). Twelve key informant interviews were conducted with four female community health volunteers (FCHVs); four health workers, the majority of which were in a leadership role (i.e., Health In-Charge, Immunization Program Manager, etc.); and four representatives of the local government (i.e., Ward Chair, Chair of the Health Facility Operation Management Committee, etc.; see Table 1). For mothers, who are the target population of the Rejoice Architecture Project, we were particularly interested in understanding the ways in which they communicate with each other about immunizations; and so, focus groups were conducted. Fifty-four mothers with children under the age of 2 participated in eight focus group discussions that ranged in size from five to nine participants (see Table 1). For two of the eight focus groups, only individuals from marginalized castes were sampled to gain a more comprehensive sense of the community's wide array of experiences.

According to Nepal's National Immunization Schedule [7], all vaccines are recommended to be administered by 15 months. Therefore, caregiving to a child under 2 years was selected as inclusion criterion for in-depth interviews and focus groups to include families at all stages of the immunization schedule. Participants were excluded if a family member had previously participated in one of the interviews or focus groups.

## Recruitment and consent

In Nepal, FCHVs are the frontline health workers at the community level. Two FCHVs from each ward assisted data collectors in recruiting potential participants for in-depth interviews

**Table 1. Description of the data collection modalities.**

| Participant | Session Type | # Sessions |
|---|---|---|
| Mothers | Focus Group Discussion | 8 ($n = 54$) |
| Fathers | In-Depth Interview | 5 |
| Grandmothers | In-Depth Interview | 5 |
| Health workers | Key Informant Interview | 4 |
| Government representatives | Key Informant Interview | 4 |
| FCHVs | Key Informant Interview | 4 |

and focus groups. Health facilities maintain an updated list of mothers with newborn children in the community. Using these lists, FCHVs identified mothers, fathers, and grandmothers of children under the age of 2 for data collectors to recruit in-person at their homes. Data collectors used convenience sampling for in-depth interviews and focus groups, using a quota system to guide their efforts. Key informants were purposively selected for having substantial involvement in immunization planning and procedures and were identified through discussions with community leaders. Key informants were recruited at their place of work by data collectors.

Informed consent was obtained from all participants immediately before data collection. Data collectors provided each participant with an information sheet and consent form, read the forms aloud to the participant, received written consent to participate, and provided the participant with an unsigned copy of the consent form. In-depth interviews with fathers and grandmothers took place in a private area in or nearby the home. Interviews with key informants were conducted in the informant's office or an empty room at their place of work. Mothers participating in focus group discussions were provided a date, time, and location for the discussion after they were recruited, often in a community center or central outdoor area.

## Data collection procedure

Data were collected in-person from January to March of 2020. Interviews and focus groups were facilitated by male and female data collectors overseen by the second, third, and fourth authors. All data collectors completed an in-person training led by the Principal Investigator which covered techniques for facilitating interviews and focus groups, probing, and conducting ethical research. Before administering the interview or focus group, data collectors identified themselves as researchers from Kathmandu interested in learning more about child immunization in Makwanpur District. To build rapport with participants and facilitate a comfortable space for sharing, data collectors began each session with two broad questions about the participant or their community. Interviews and focus groups lasted approximately 60–90 minutes and were audio-recorded. During all interviews and focus group discussions, two data collectors were in attendance; one to facilitate the session, the other to take detailed notes. The data collector that was not facilitating the conversation recorded field notes to document their perceptions of the participants' attitudes and willingness to engage, as well as the surroundings, interruptions, or other notable characteristics of the session.

In one case, a key informant did not consent to audio-recording, and so, field notes were used to document this participant's responses. During two focus group discussions, one mother in each group left the discussion early to tend to her crying child. Aside from these instances, no participants dropped out or refused to participate. No repeat sessions were conducted.

Audio-recordings of interviews and focus group discussions were translated from Nepali to English, and transcribed verbatim by native Nepali-speaking team members (see Paul et al. [19] for data management procedures). Through this process, conducted in phases as the data were collected, the researchers observed data saturation by the time data were collected in the fifth catchment area. Although data collection was initially planned for a sixth catchment area, no additional data were collected. Additionally, participants were not asked to comment on or correct transcripts, as native Nepali study team members with extensive experience working with communities provided context and clarity.

## Instruments

Specific topics covered in each interview and focus group varied based on the instrument in use and the flow of the conversation. Each instrument was designed to assess topics of

immunization facilitators, barriers, and social norms (see Paul [24] for the instruments used). In addition, the in-depth interview guide covered items related to the family's experiences with vaccination and family involvement in child health and decision-making. The key informant interview guide included items related to immunization coverage in the ward, typical communication between caregivers and health care providers, the physical environment of the local health facility, and opinions of their workload. For the focus group guide, items regarding characteristics of an ideal health facility, a description of their local health facility, and typical communication between caregivers and health care providers were also included. All instruments were designed collaboratively by study team members from both the US and Nepal to assure cultural appropriateness. Instruments were translated from English to Nepali, pre-tested in a non-study site in Makwanpur, and revised for comprehension and cultural congruence.

## Data analysis

Four study team members analyzed the data using an iterative, grounded theory approach informed by Glaser [25] to uncover emergent themes. Each researcher held, or was working toward, a relevant graduate-level degree and had experience leading or analyzing qualitative research related to maternal health, infant and child health, health services, or other health topics in developing settings.

The researchers began by independently reading through the transcripts to gain familiarity with the data, then met to identify prominent patterns and develop the initial codebook. Researcher triangulation was then used to ensure the credibility of analyses. Two researchers were assigned to each transcript for coding. Researcher assignments rotated on a weekly schedule to ensure researchers were working within different pairs throughout the analysis process. Each researcher then coded their assigned transcripts independently, using NVivo (released in March 2020). Research pairs met at least once per week to compare codes, discuss discrepancies, and reach a consensus for each transcript. The full analysis team also met weekly to revise the codebook. After all transcripts were coded, the team reviewed previously coded transcripts to align with the final codebook.

Following coding, prominent themes were identified by comparing codes across the various types of interviews; running word queries; and creating visual conceptual maps to link patterns. Researchers thematized both the demand-side (mothers, fathers, grandmothers) and supply-side (FCHVs, health workers, Palika representatives) facilitators and barriers of immunization as well as the connection between the two.

## Results

From the analyses, three major themes emerged, including demand-side facilitators of and barriers to immunization, supply-side facilitators of and barriers to immunization, and lack of mutual trust between the demand- and supply-sides. See Fig 1 for a map of all themes and sub-themes.

### Demand-side facilitators of and barriers to immunization

**Facilitators.**    Results of interviews and focus groups indicate high demand for vaccines in Makwanpur. Mothers repeatedly emphasized the benefits of immunization and their desire to protect their children. In fact, these positive perceptions of immunization seemed to translate to actual behavior (or at least the perception of immunization behaviors), as the majority of participating mothers responded that nearly all children in their communities have been immunized. High demand for vaccines among mothers of young children appeared to be

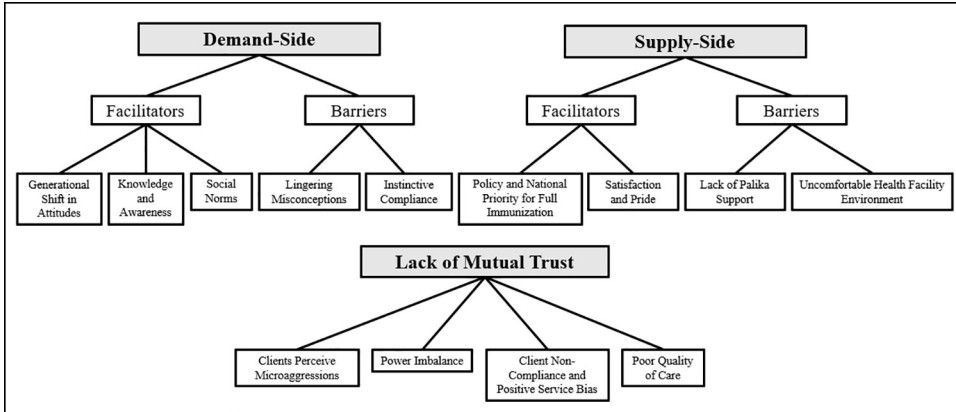

**Fig 1. Map of themes and sub-themes.**

facilitated by several factors, including a generational shift in attitudes toward immunization, notable awareness and knowledge of immunizations, and positive social norms.

Participants shared that members of past generations, such as parents or grandparents, were often fearful and untrusting of vaccines. Mothers and family members expressed that attitudes are different today. Generally speaking, mothers now understand the need for vaccines and weigh the benefits of having a healthy baby over the costs of minor side effects. For example,

> In the past, people didn't used to go to get immunized as the child would cry and have fever. Now, mothers take their children, even though the child cries and gets fever as well. We all have immunized our child. (Mother)

When asked why immunization has increased in their community, respondents often cited education as the main determinant. As one respondent put it,

> Maybe because mothers are educated now. . .Because we think it would be better for the children. Even though we didn't get the vaccines when we were children, we wish that would not happen to our children now. So, everyone brings their children. (Mother)

In addition to a shift in attitudes, the demand for vaccines may also be facilitated by growth in awareness and knowledge among mothers. Focus group discussions revealed that mothers in Makwanpur are fairly knowledgeable and have an understanding of vaccines greater than prior literature might suggest [12, 15, 18, 26]. Most participants were able to articulate the purpose of vaccines and describe the common side effects.

This heightened sense of awareness and knowledge is facilitated in numerous ways. Some caregivers attributed their knowledge to the receipt of education. Mothers also identified other sources of information, most commonly FCHVs who counsel mothers to "ensure that no child miss their vaccines." Mothers also frequently reported learning about immunization through social networks. For example,

> We also share the information among our friends. I tell her what I know, she tells me what she knows. So, there is no one who would not know. Everyone gets the information. (Mother)

Some mothers also reported learning about vaccines and vaccine scheduling from their health workers during pregnancy or well-child visits. However, this finding was not consistent across all focus groups.

Because of the generation shift and increased awareness in the community about the importance of vaccines, immunization has become the norm. It was widely reiterated by mothers and stakeholders alike that almost everyone takes their children for vaccination and that people in the community encourage and support each other to vaccinate their children.

> There is no one in our community till now [who does not vaccinate] . . . Everyone gets full immunization here. . . We all have immunized our children. (Mother)

Indeed, quotes revealed that several mothers served as vaccination ambassadors, sharing information about vaccines with their peers and educating the community.

> It is well known that one should vaccinate. We tell the people in the community. Also, the FCHV tells us in meetings. (Mother)

**Barriers.** Although we observed high demand for immunizations, marked by several facilitating factors, we identified a few barriers to sustaining vaccine uptake in these communities. These barriers included lingering misconceptions and negative attitudes toward immunization in the community and among family members as well as an instinctive tendency for some mothers to comply with health worker recommendations at the expense of their own learning and understanding.

Despite progress made by younger generations, results indicate that remnants of more negative attitudes toward vaccines still exist. For some mothers, pressure to forgo immunization and other health services may even come from their own families.

> It is also due to old beliefs in people as well. There are people who mention that in the past, we didn't immunize, so why should you get immunized. There are also such kind of mother-in-law as well, who say such to their daughter-in-law. Even when the daughter-in-law wants to go for ANC checkup, the mother-in-law will say, "We never went, why would you want to go as well?" (Grandmother)

As depicted in the quote above, most of the comments made about negative attitudes from older generations were perceptions of others' (not participants' own) parents, grandparents, and in-laws. Therefore, this theme may reflect the projection of community norms rather than personal beliefs.

In addition, some caregivers reported that they bring their children for vaccination because they are prescribed to do so by the health workers and FCHVs. These parents reported getting their children vaccinated because they are supposed to, and it is expected of them, despite not being fully aware of the exact benefits.

> We get information from the health workers and FCHV. They tell us [to immunize our children] for our own good. Vaccines are good for our children, so we take them for vaccinations. (Mother)

> We get information from the health workers and FCHV. We understand and trust what they say. I don't think there is disadvantage of vaccines. I don't know if it has any. (Father)

## Supply-side facilitators of and barriers to immunization

**Facilitators.** Interviews with key informants revealed that the major factors facilitating the supply of vaccines are the national priority of and policies related to immunization in Nepal and satisfaction and pride among health providers.

According to service providers, the supply of vaccines is largely affected by the messaging from Nepal's government. In response to the "Reaching Every Child" initiative of the government to achieve full immunization, Palikas have been implementing local immunization programs. Reports from this sample indicate that Palikas have conducted awareness campaigns, home visits, and other initiatives to ensure no child misses vaccination.

> During 2073–2074 (2016–2017), there were cases where people didn't want to get vaccinated. Then, we conducted door-to-door service. We provided information and made people aware. Once we had done that, we kept them on record in our registry and followed up with them. We continued with that process and now there is no such problem. Everyone comes here. (Health worker)

Health workers and FCHVs also described personal satisfaction and a sense of pride that comes from contributing to the community, which motivates them in their jobs. Some expressed how their daily work even provides them with relief from stress in their personal lives.

> I feel enjoyable as we have been able to provide good services and people also are satisfied with our services. We also forget about the tensions at home when we come here. (Health worker)

In addition, service providers expressed confidence in their ability to efficiently perform their duties and expand upon the services offered at health centers. Self-confidence was observed to be a key motivator for providing quality services, including immunization.

> When I had come here, there was nothing here. We started the birthing center. We have initiated two immunization centers. We also established a lab, and it has been successful. (Health worker)

**Barriers.** Despite these facilitating factors, supply of immunization is hindered by several factors in Makwanpur. Barriers to immunization include diminishing will of and support from Palikas to promote immunization and uncomfortable health facility environments.

Although the Nepal government has achieved the implementation of local immunization programs from the 'Reaching Every Child' initiative, reports from these interviews suggest that the resulting effort from Palikas may be short-lived. At several study sites, participants reported that immunization activities diminished after full immunization was achieved once, and progress was not sustained. As one participant put it,

> There were awareness programs before, but in recent times we haven't held any awareness program or campaign for increasing the coverage of immunization here. (Palika representative)

Health workers expressed that immunization efforts are further hindered by inadequate logistics for other health services from Palikas. The frequent shortage of health resources demotivates service providers and highlights a weakness in supply chain management in the

broader health sector. Moreover, participants, particularly FCHVs, disclosed a shortcoming in staff training. It was well understood that providers should have sufficient knowledge and skills to understand the most recent developments in the health sector and provide care accordingly. Yet, the need for more frequent staff training was felt widely by the service providers in this sample.

Health service providers also expressed dissatisfaction towards the amount of support they receive from Palika authorities, stating that there is a lack of health facility monitoring, which leads problems to go unnoticed and unresolved. Additionally, some health workers conveyed a fear to voice their concerns to authorities, while others chose to confront the Palikas despite the possibility of being labeled as a troublemaker.

> Despite informing the ward chair and other members 4–5 days prior to the programs, like Measles-Rubella campaign, [authorities] never show up in the programs. Only FCHVs attend the programs even though they have to travel from far. I want to raise this issue to the Palika, but I fear that I will be accused of setting the wrong impression. However, I am thinking of raising this issue in our own meeting, as I am not satisfied with the behavior of the Palika staff. (Health worker)

Results indicate that the physical and social environment of health facilities can be an important motivator for service providers; but when lacking, can impede the service delivery process. Participants discussed the need to improve the infrastructure of the health facilities, cleanliness of the internal and external environments, hygiene and sanitation, overcrowding, and the amount of time allotted for consultations in order to provide high quality care, both in terms of immunization and general service delivery. Participants reported that while they have managed with meager conditions and limited resources, there is a need for improvement in the overall environment of the health facilities.

> The cleanliness is okay, sir. The health post has cracked in some parts. But compared to how it was in the past, it is better. It is not crowded. But the cleanliness is not very good. (Health worker)

## Lack of mutual trust

Throughout the data, themes of mistrust and lack of mutual respect between health workers and clients were pervasive and appeared to partially explain the context of existing service gaps, including those related to immunization uptake and continuation. There appeared to be a bi-directional relationship in terms of trust; one direction in which, in the eyes of the clients, mistrust is generated by health workers towards them, and the other in which, in the eyes of the health workers, mistrust is generated by clients, particularly those from marginalized and disadvantaged groups. Lack of mutual trust was presented through four key illustrations, including 1) micro-aggressions, 2) power imbalance, 3) poor quality of care, and 4) client non-compliance and provider bias.

**Clients perceive micro-aggressions.** Poor facility management and unwelcoming physical environments fostered feelings of discrimination among clients. This was expressed in instances in which mothers were asked to sit on the floor in the absence of a bench or chair in the waiting area. In other cases, mothers felt that they were asked to wait longer for consultations because of their identity or education status. As one mother shared,

> The sisters (health workers) do what the clever people in the community tell them to do. Don't they? It's like that. Some can speak [the local dialect], some cannot. They should

work on understanding people and treat all equally. But the sisters, doctors don't understand. (Mother)

Participants reported that they felt inferior for being treated poorly by health workers and suggested that staff improve their behavior and speak more politely. One key informant surmised that disrespectful health worker behavior may be a result of irritation caused by crowded health facilities, particularly on designated immunization days.

In many cases, clients opted for experiences at private health facilities, despite having to pay expensive fees. In comparison, public facilities provide all services for free. Some clients have come to the understanding that when one pays for a service at a private facility, the service provider will be held more accountable for their actions. From their perspective, it is the responsibility of the private provider to retain the client by ensuring their experience is good. That expectation is not maintained at public facilities.

**Power imbalance between health workers and clients.** In our data, there was a distinct display of power imbalance between health workers and clients within the health facilities. This became evident as mothers described being reluctant to ask questions or have extended conversations with health workers during consultations or vaccination appointments. Some mothers reasoned that this hesitancy derived from being "ignorant" and "uneducated," while others suggested that health workers should be held accountable for their actions toward clients.

We don't ask even though we have questions (laughter). We find it difficult. We feel awkward to speak (laughter). We are scared because we are not educated and don't know much. (Mother)

**Poor quality of care.** In general, mistrust was not directed at one health facility in particular, but at the broader health system in Nepal. Across focus groups, mothers expressed that public health facilities are incapable of delivering comprehensive and effective care. Mothers also reported that public facilities do not have essential medicines, equipment, or human resources, and are not open enough hours during the day to adequately serve the community.

There are no medicines for pneumonia, typhoid. There is no medicine for cough, let alone pneumonia. No medicines for rashes, injuries. The [auxiliary nurse midwives] tell us to go to the city because those medicines are not available there. If anything happens, they show us the medical (private facility). They give us paracetamol for fever, that's all. Paracetamol for everything, fever, headache. (Mother)

In addition, mothers explained that health workers are not fully skilled because they prescribe medications that are "low quality," "not-so-strong," or "expired." As one mother put it, "Health workers should be capable of prescribing appropriate medicine, which they are not." This was not only reported by mothers and other clients, but by local officials as well. Some Palika representatives expressed a belief that the government procures medicines from local vendors which intentionally supply low-quality products. It is important to note that all pharmaceutical companies are required to fulfill quality standards before they supply medicines.

**Client non-compliance and positive service bias.** According to health workers, clients can also cultivate feelings of mistrust through non-compliance. Many health workers reported the expectation of clients to abide by certain health behaviors, such as giving birth at the health facility rather than at home or bringing their children for immunization by the national

schedule. However, when clients did not meet these expectations, health workers reported perceiving them as unreliable or irresponsible.

Throughout the data, a pattern emerged among health workers to encourage client compliance that we note as *positive service bias*. Positive service bias refers to situations in which health workers treat clients who comply with service uptake at health facilities more preferably than those who do not. Such services most commonly include institutional delivery and child immunization. For example, some participants reported that women who do not comply with these expectations are deprived of services or treated poorly.

> In our village, they tell us that those who are not born in health facility or hospital are not provided vaccines. (Mother)

> I feel that they care for those, talk to those who have delivered there, for both mother and child. . . and they don't care for those who deliver at home. (Mother)

It became apparent that some health workers were using ethically questionable techniques to persuade mothers to adopt these practices. In some cases, health workers threatened mothers with legal consequences (a power which they did not hold) if they did not immunize their child. Indeed, one health worker even talked about this practice without compunction,

> We also create a kind of fear in mothers saying, "If you don't receive the full immunization card, then your child will not get his citizenship. Your child can't go to a foreign country. So, in order to get this full immunization card, he must take the immunization till 15 months." We create such an environment. . . so that they come to get immunization. (Health worker)

It is important to note that refusing citizenship on the condition of full immunization is not a law practiced in Nepal, and this is falsely being used to create fear by this health worker.

## Discussion

A key objective of this research was to gain a better understanding of the barriers and facilitators for improved immunization. We found some of these barriers and facilitators pertained to caregivers, others to providers, and still others to these two parties jointly. We elaborate on each of these categories below, making recommendations for how these findings could inform interventions.

A notable finding from our work was the relatively high levels of immunization-related awareness, knowledge, and acceptance among caregivers. To some extent, this may reflect the higher rate of overall immunization rates in the Makwanpur district (68%), as compared to the country's average of 65% immunization coverage [3], and the barriers and facilitators we have identified likely speak to the last "tail" of the curve in Nepal's pursuit of full coverage. We found that existing social norms were highly supportive of immunization, and many respondents acknowledged that this represented a generational shift. According to participants, girls are receiving more education today than their elders have in the past, supplying them with the knowledge to make informed decisions for their and their children's health. In fact, young mothers today are demonstrating skills in advocacy and health communication within their families and communities to encourage greater vaccine uptake, dispel myths, and alleviate lingering fears.

Despite this encouraging progress in mothers' understanding of and attitudes toward immunization, the process of translating this into actual behavior is at risk due to several

barriers. Caregivers are still facing opposing views from elders in their communities and, in some cases, their own families, who discourage them from vaccinating their children because they themselves were not vaccinated and have remained in good health. Depending on the level of autonomy a woman has in her family over the decision-making for her child's health, these lingering misconceptions may impede her ability to vaccinate her child. In addition, although many mothers in this sample displayed adequate knowledge of vaccines to support their favorable attitudes toward immunization, several demonstrated a tendency to comply with health worker recommendations without fully understanding the reasoning behind them. This impulse to act without asking questions threatens the sustainability of vaccine uptake as these caregivers may be more susceptible to vaccine misinformation.

Providers themselves were also battling many issues, one of which was the lack of support they reported receiving from the Palika authorities and others in charge. Providers noted the relative absence of monitoring of their efforts, not being heard, and not having received ongoing professional training. While working under these challenging conditions appears to add to their professional stress, many were, nonetheless, passionate in their desire to raise the ceiling on overall vaccination rates. Indeed, their motivations for improving vaccination rates appear to have led some to display highly disapproving and even discriminatory practices toward caregivers who missed their infants' vaccinations.

## Positive service bias

Perhaps a key finding from this research is our observation of how certain segments of the population are being left behind because of the pressures exerted in the system to reach 100 percent compliance. Ironically, the government's priority for full vaccination has pushed some clients out of the health service delivery chain for two primary reasons. First, various researchers have noted that clients who miss vaccinations are likely to be members of the most vulnerable and marginalized groups [27–29]. They face significant distance and travel challenges and often cannot allocate time for travel, mostly because of financial constraints and the need to be continuously engaged in income-generating tasks. Members of these groups are likely to miss vaccines, despite their better knowledge and desires. Once they miss a few vaccines, they become fearful to visit the health facility to catch up with missed regimens due to fears of how they will be treated by the providers.

The second part of this equation pertains to our earlier point about providers' zeal for full vaccination coverage that often drives them to treat non-complying caregivers with disrespect and even contempt. Indeed, we saw this play out in how providers discussed those who did not comply. This kind of behavior was also prominently noticed by caregivers, who related to us the maltreatment they received from providers.

This fear of being ostracized, and being implicitly labeled as an irresponsible parent, made it more likely that they would discontinue their vaccinations altogether. This is what we term a *positive service bias*, the idea that providers, acting with good intentions to maximize vaccination rates in their communities, inadvertently drive caregivers away when caregivers have been occasionally non-compliant. Positive service bias may be conducted through social interactions (i.e., ignoring or speaking rudely) or service removal (i.e., threatening citizenship or school admission), which may further discourage caregivers from seeking health care from the health facilities.

## Recommendations

The intervention implications of our findings from the demand side–that immunization attitudes are already substantive but that several barriers threaten the sustainability of

immunization behaviors–is two-fold. First, immunization interventions should focus not on persuasion (to vaccinate), as rates are already high and attitudes favorable, but rather on the facilitation of people's desires to immunize their children. Instead of trying to convince people to vaccinate their children, efforts should address barriers so that people can translate their intentions into behaviors. Some of those barriers are physical and structural, while others are interactional, pertaining to how providers and caregivers interact with each other.

Second, health communication messages should focus on addressing normative mispercep-tions about the prevalence of vaccination rates, by highlighting the fact that full immunization is the norm, not the exception, in most places. Indeed, given the high immunization rates, this type of messaging could be based solely on making people aware that most of their neighbors vaccinate their children and that those who do not do so constitute only a minority. These efforts could also model the ways in which caregivers have overcome specific barriers to vacci-nation and provide support systems based on caregivers assisting each other–by, for example, pooling childcare responsibilities so that mothers with multiple children are able to travel with the infant needing vaccinations.

Our findings from the supply side–that health workers can be highly motivated to pro-vide quality services but require adequate support from Palikas to do so–speaks to the need to provide appropriately tailored training for providers. More frequent training could help providers understand the plights faced by caregivers that might account for why they have missed vaccinations, and also assist providers in adopting a compassionate, rather than a punitive, approach. The health system also needs to address this positivity bias, and one way of doing so may be to allow for, indeed incentivize, "catch-up protocols," procedures put into place specifically for helping close the gap on prior vaccines that may have been missed.

Lastly, we point to the need to focus on building trust in the health system. The public health sector runs not on a market-based system, as all services are provided free of charge, but on trust and mutual respect. Caregivers are more likely to use vaccination services if they feel respected by providers and the overall health system and if they can trust the quality of care they receive. Nepal is relatively close to achieving full vaccination coverage, and it appears that this last stretch requires work on both the demand- and supply-sides. On the demand-side, greater trust is required on the part of marginalized groups that have been left out in other major health initiatives. But such trust can come only if these groups believe that the supply-side system is invested in and acting on their interest, an indication of which may be a feeling of being treated with respect when they visit health clinics.

## Limitations

Because this study was conducted in one district of Nepal, the experiences of health workers and community members we report may vary from those in other parts of the country. Our study district, Makwanpur, has a higher immunization rate (68%) than the national average (65%), which may explain the positive social norms and high demand related to immunization depicted in this study [3]. Additionally, Makwanpur is geographically more accessible from Kathmandu, with more health facilities and immunization centers than other more remote districts. Hence, people's access to care and enhanced transportation facilities (at least com-pared to many remote districts in Nepal) render our findings a bit less generalizable and some-what privileged. Another limitation is that we did not perform member-checks in that the transcripts were not reviewed by participants. This oversight could introduce some error in our interpretations, but we do note that interviews were captured by both recordings and simultaneous notetaking by a second researcher during the interview.

We also note that participants in the study were limited to those who lived within a one-kilometer radius from the clinic. This means that study findings may not generalize to others at greater distances (and thus facing additional barriers).

## Conclusion

Significant hurdles remain in Nepal's ability to meet its immunization goals. Some of the barriers reside on the demand-side (lingering negative attitudes in the community, instinctive tendency to comply without full understanding), some on the supply-side (diminishing will of and support from Palikas to promote immunization, uncomfortable health facility environments), and others at the healthcare clinic itself, which sits at the nexus of supply and demand (lack of mutual trust between caregivers and health workers). Immunization can likely be further enhanced when caregivers' interactions with health care providers in the clinic and providers' interactions with caregivers are based on mutual trust and respect. This can be done through proper training and awareness-raising campaigns, but it can also be significantly improved when the physical and infrastructural environment in the clinic itself improves.

We conclude that in communities where attitudes toward immunization are favorable, future interventions should transition from educating and persuading parents to immunize their children to focusing on removing social and structural barriers. We offer a variety of recommendations to mitigate these barriers, including providing tailored training for providers, incorporating behavior-change approaches to promote compassion over penalty, and implementing "catch-up protocols" to encourage communities to continue striving for full immunization despite setbacks. Physical and structural barriers are also present, namely neglected infrastructure and inadequate medical supplies. These findings suggest that investing in the quality of health facilities, such as providing higher quality supplies and designing a welcoming physical space may remove barriers to seeking care and encourage greater uptake of services.

Our most prominent conclusion, however, is the influence of trust between health workers and caregivers on immunization and other service uptake. Behaviors of providers depicting a positive service bias (whereby they treat more positively patients who have complied, compared to those who have not) are creating a cycle of mistrust as health workers condemn clients for "deviant" behavior and affected individuals are discouraged from partaking in future health care. The process of building trust between the demand- and supply-sides is bi-directional, requiring effort from both groups to listen to and acknowledge each other's needs and experiences. Although it begins on an interpersonal level, we believe that building trust within Nepal's national health system may be the key factor to achieving the last stretch to full immunization. This is certainly worthy of future inquiry.

## Supporting information

**S1 File. Mother FGD guide.**
(DOCX)

**S2 File. Grandmother interview guide.**
(DOCX)

**S3 File. Health worker interview guide.**
(DOCX)

**S4 File. Palika representative interview guide.**
(DOCX)

**S5 File. FCHV interview guide.**
(DOCX)

**S6 File. Father interview guide.**
(DOCX)

## Acknowledgments

We thank the Thaha, Bakaiya, and Kailash Municipalities for their collaboration. We also express gratitude to the staff and leadership at each participating health facility for their partnership and making this research possible.

## Author Contributions

**Conceptualization:** Rajiv N. Rimal.

**Data curation:** Shraddha Nepal, Kamana Upreti.

**Formal analysis:** Alicia M. Paul, Shraddha Nepal, Kamana Upreti, Jeevan Lohani.

**Funding acquisition:** Rajiv N. Rimal.

**Investigation:** Shraddha Nepal, Kamana Upreti.

**Methodology:** Alicia M. Paul, Shraddha Nepal, Kamana Upreti, Jeevan Lohani, Rajiv N. Rimal.

**Project administration:** Alicia M. Paul, Shraddha Nepal, Kamana Upreti.

**Resources:** Alicia M. Paul, Shraddha Nepal.

**Supervision:** Alicia M. Paul, Jeevan Lohani, Rajiv N. Rimal.

**Validation:** Jeevan Lohani, Rajiv N. Rimal.

**Visualization:** Alicia M. Paul.

**Writing – original draft:** Alicia M. Paul, Shraddha Nepal, Kamana Upreti, Jeevan Lohani, Rajiv N. Rimal.

**Writing – review & editing:** Alicia M. Paul, Rajiv N. Rimal.

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
