## [Decision Letter · Decision Letter 0]

9 Sep 2021

PONE-D-21-16867The last stretch: Barriers to and facilitators of full immunization in NepalPLOS ONE

Dear Dr. Paul,

Thank you for submitting your manuscript to PLOS ONE. After careful consideration, we feel that it has merit but does not fully meet PLOS ONE’s publication criteria as it currently stands. Therefore, we invite you to submit a revised version of the manuscript that addresses the points raised during the review process.

We look forward to receiving your revised manuscript.

Kind regards,

Andy Stergachis, PhD

Academic Editor

PLOS ONE

Journal Requirements:

Additional Editor Comments (if provided):

Title: Missing is the target group, i.e., 'in children'. Consider adding to the title.

Methods: Provide additional information on the characteristics of the geographical location of the study, such as population demographics and health burden, particularly in young children.

Results: Why do some subheadings not include quotes from participants while most of the others do?

Discussion: Strive to better organize as well as shorten its length.

Limitations: Additional limitations are cited by a reviewer.

Reviewers' comments:

Reviewer's Responses to Questions

**Comments to the Author**

1. Is the manuscript technically sound, and do the data support the conclusions?

Reviewer #1: Partly

Reviewer #2: Yes

2. Has the statistical analysis been performed appropriately and rigorously? 

Reviewer #1: N/A

Reviewer #2: Yes

3. Have the authors made all data underlying the findings in their manuscript fully available?

Reviewer #1: Yes

Reviewer #2: Yes

4. Is the manuscript presented in an intelligible fashion and written in standard English?

Reviewer #1: Yes

Reviewer #2: Yes

5. Review Comments to the Author

Reviewer #1: Thank you for the interesting work done. Kindly please refer to my suggestions below:

General points

It would be appropriate to separately summarise the findings of quantitative studies conducted in Nepal related to the topic in the introduction section. A reasonable rationale for conducting this study is required.

The objective of this study is not clearly mentioned. Under the objective, it says the authors sought to understand the implication of the relationship between service seekers and health workers as well as the influence of clinic environment in immunisation service uptake. Next, it is stated that it would like to address barriers at the nexus of both demand and supply sides. Finally, it has also talked about the aim of the Rejoice Architecture Project. However, the abstract of this study stated a very specific objective – to understand the barriers to and facilitators of immunisation in Makwanpur from both the demand and supply sides.

Then coming to the result section, it is again not clear what are the themes actually about. Understandably, barriers and facilitators can be included under the same theme under the demand and supply side. However, themes are not reflective of results most often. For instance, while the first theme looks like it only tries to include the facilitators of immunisation uptake, perspectives from both the demand and supply sides are included. If it is to be presented this way, then the authors should not confuse the demand side with the quote from health workers (as the authors mentioned in lines 230 to 231 that demand side of data is collected from family members' perspectives and supply-side from health workers or representatives' perspectives). Alternatively, based on the title of this study, separate headings on facilitators and barriers and themes within those headings could be done.

Overall, there appeared to be a poorly consistent flow through the objectives, results, discussion, and conclusion. The discussion and conclusion section should be coherently organised as per the objective of the study and rewritten.

Also, the authors should mention why there are no quotes from fathers at all. Was the information not at all noteworthy? It should be explained in the manuscript.

More detailed comments:

Introduction

Line 59 – NIP, in 1977, was introduced as an immunisation program in the country, not to address inequities in vaccine coverage.

Line 84 – reference number 13 is not appropriate to cite here as it was carried out in Kenya. Also, reference 12 focused on vaccines scheduled to be completed by 12 months of age. Would you please cite the appropriate reference here?

Lines 115 to 116: "In Nepal, health facilities are often crowded, have inadequate waiting spaces if any at all, and are often unclean." It is such a generalised sentence. It should either have a precise citation or should be removed.

Methodology

Although a qualitative method is stated, no specific study design has been mentioned. Would you please keep the heading study design and mention it?

Lines 131 to 132 – Being accessible from Kathmandu doesn't make it suitable for the research. The other two reasons are quite fair.

Line 135 – I wonder why individuals living within a 1 km radius of health facilities were only eligible participants as the distance to health facilities could be a barrier, especially in hilly regions of Nepal.

Line 138 - Are there any reasons why mothers were not included in in-depth interviews?

Line 172 – Data collection procedure?? Also, please mention the data collection period as it could be valuable data.

Lines 190 to 192 – Could you please associate the reference cited with this study? If not, you need to mention the procedure here clearly.

Lines 195 to 197– Providing enough context and clarity doesn't justify why member checking was not done to increase the credibility of the research.

Line 199 – Does it mean each interview and focus group discussion covered a separate topic? If so, it limits the transferability of the research. Can you please explain?

Line 200 – The instruments should only be a guideline. It is not clear whether there was a separate guideline for each topic. Also, please include interview guidelines as an additional file.

Results

Line 307 - Here, the gender norm of women being the primary caretaker does not justify being a facilitating factor to increase vaccine coverage. Instead, the link between women receiving immunisation awareness when participating in mothers' groups and being the child's primary caretaker increased the likelihood of vaccinating their child.

Line 344 – The last paragraph within this sub-theme is more of accountability issue and mistrust as represented by the theme, not a power imbalance.

Line 364 – Is 5-hours a regular office opening hours, or was it because of the absence of health workers in the facilities to provide service. Immunisation has been covered mainly as an outreach service in communities in Nepal. Does the above-mentioned service hour refer to it? Would you please address this clearly?

Line 380 to 383 – It is not acceptable to include information from the researcher's perspective in the result section, especially in this type of study. If it is explained based on the participants' response, please remove "Generally speaking" and support this claim with participants' quote.

Line 384 to 389 – This is going out of context unless linked with the study's objective. Are they talking about addressing adverse effects of immunisation or the mistrust due to previous experiences on immunising their child? Or are they just expressing their other concerns related to health care services?

Line 394 – This sentence does not reflect the essence of the quote here. It is more about not getting other services if not fully vaccinate their child. Refusing to immunise a child by health worker is a sensitive claim, and therefore a quote is required.

Line 431 – Why family planning devices is affecting immunisation efforts?

Line 445 to 453 – Why the results are not linked with immunisation here? It looks more like an interview related to the condition of the health care facility rather than how the health facility environment affects vaccine uptake.

Line 454 – Provider or Provider’s

Discussion

Line 518 – Time hasn't been explained clearly as a barrier in the result section in the way it is mentioned here in the discussion.

Line 524 to 526 – It is expressed from the key informants on the supply side rather than the family caregiver in the result section. It is misleading.

Overall, the discussion should be coherently organised as per the objective of the study and rewritten.

Limitations

Line 588 to 589 – "It is also a fertile district able to grow and export cash crops to neighbouring cities of 589 Kathmandu and Hetauda" – This sentence has nothing to do with this study unless the socio-economic status of study participants is linked with vaccine uptake.

There could be more limitation

– one could be member checking (as mentioned earlier)

- Including participants within 1 km radius excluded participants who could be affected by travel time to health facilities

- No information is provided about the participants' backgrounds. It would be worthy of increasing the transferability of the study findings.

- Was the number of participants finalised beforehand? Or was it based on data saturation?

Conclusion

Line 594 – "stocking adequate doses of vaccines" - not mentioned in the result section and, therefore, cannot be included in the conclusion

Line 595 – "people's perceptions that many others do not fully vaccinate their children" – How is this a barrier? And why is it on conclusion rather than other important barriers discussed in the result section?

Line 596 – "feelings of experienced disrespect" – What does this mean?

Line 608 – There's nowhere in the results section that travel time is a barrier.

Line 617 – Who are these marginalised groups? There is no identification of marginalised groups' deviant behaviours in the results.

Overall, the conclusion should be rewritten.

Reviewer #2: The manuscript is overall well written, based on findings from the field in one district of Nepal. This is recent work and as such it showcases situation of immunization after Nepal adopted provincial system of governance in 2015. This study is one of a very few detailed assessments of state of immunization in Nepal in recent years.

The sample size is somewhat an issue, given the information that the work is trying to capture. The one district (out of 77 in the country) that was chosen is in close proximity to the capital of Nepal, Kathmandu and as such it may not truly represent the situation prevalent in rest of Nepal. Further, only three palikas were chosen with very limited sample size. The authors should make sure that this is identified as a major limitation of the study, as the current title of the study and its objectives do not match the results provided. Perhaps the title and objectives of the study should be revised to truly reflect the work carried out.

6. PLOS authors have the option to publish the peer review history of their article (what does this mean?). If published, this will include your full peer review and any attached files.

Reviewer #1: No

Reviewer #2: **Yes: **Sameer M Dixit

---

## [Author Response · Author response to Decision Letter 0]

8 Nov 2021

Response to Reviewers

We are grateful to the reviewers and editors for their comments. We have addressed all of them, as noted below.

Response to Editor Comments:

We have reviewed PLOS ONE’s style guidelines and have made the appropriate adjustments. If there are specific formatting requirements we have missed, please let us know and we will happily address them. 

Title: Missing is the target group, i.e., 'in children'. Consider adding to the title.

We have revised the title to include the target group (children) and specify the district in Nepal where the study was conducted (Makwanpur). 

Methods: Provide additional information on the characteristics of the geographical location of the study, such as population demographics and health burden, particularly in young children.

We have added information to the Study Setting section on Makwanpur’s population size, land area, geographic diversity, ethnic diversity, and child immunization rate. Other health indicators are often reported at the Regional level in Nepal, and so District-level data are not readily available. 

Results: Why do some subheadings not include quotes from participants while most of the others do?

All subheadings now include quotes from participants.

Discussion: Strive to better organize as well as shorten its length.

The discussion section has been reorganized to follow the organization of the results section: demand-side factors, supply-side factors, and issues of trust at the intersection. Then, we have grouped our intervention implications into one sub-section labeled “Recommendations”. We have also cut down on the length by focusing only on the most prominent takeaways and conclusions. We hope this will help to flow more coherently for the reader. 

Response to Reviewer 1’s Comments: 

It would be appropriate to separately summarise the findings of quantitative studies conducted in Nepal related to the topic in the introduction section.

Due to the limited number of studies conducted among our target population in Nepal, specifically, we feel it is best to present the information together rather than separated out by study design. 

The objective of this study is not clearly mentioned. A reasonable rationale for conducting this study is required.

To improve clarity of our study’s objectives, we have removed the sentences elaborating on our objectives and those of the umbrella study, the Rejoice Architecture Project. We hope by simplifying our objective statement and keeping it consistent with the abstract, the aims of the manuscript will be clearer. Additionally, we have stated that part of the objective of this study is to inform the intervention design for the Rejoice Architecture Project as justification for conducting the study. 

Then coming to the result section, it is again not clear what are the themes actually about… themes are not reflective of results most often… the authors should not confuse the demand side with the quote from health workers… based on the title of this study, separate headings on facilitators and barriers and themes within those headings could be done.

The themes have been re-organized to align with the objective section. They are now broken down into 1. Demand-side facilitators & barriers, 2. Supply-side facilitators and barriers, and 3. Lack of mutual trust between the demand- and supply-sides. 

Overall, there appeared to be a poorly consistent flow through the objectives, results, discussion, and conclusion. The discussion and conclusion section should be coherently organised as per the objective of the study and rewritten.

We have reorganized the results and discussion sections based on the reviewers’ feedback to follow the structure of our objectives section. We hope this resolves the issue of flow throughout the paper. 

Also, the authors should mention why there are no quotes from fathers at all. Was the information not at all noteworthy? It should be explained in the manuscript.

Quotes from fathers largely reiterated the themes presented by the mothers/caregivers. We felt quotes from mothers would be stronger, as they were the target population on the demand-side. However, we have added a quote from a father under demand-side barriers. 

Introduction

Line 59 – NIP, in 1977, was introduced as an immunisation program in the country, not to address inequities in vaccine coverage. 

This statement has been removed. 

Line 84 – reference number 13 is not appropriate to cite here as it was carried out in Kenya. Also, reference 12 focused on vaccines scheduled to be completed by 12 months of age. Would you please cite the appropriate reference here?

Our literature review is not limited to studies conducted in Nepal. Although we aim to highlight these studies as much as possible, we include references to studies conducted in other low- and middle-income countries which have similar barriers and facilitators to immunization as those in Nepal. 

Lines 115 to 116: "In Nepal, health facilities are often crowded, have inadequate waiting spaces if any at all, and are often unclean." It is such a generalised sentence. It should either have a precise citation or should be removed.

This sentence has been removed.

Methods

Although a qualitative method is stated, no specific study design has been mentioned. Would you please keep the heading study design and mention it?

We have specified that we used a grounded theory design. 

Lines 131 to 132 – Being accessible from Kathmandu doesn't make it suitable for the research. The other two reasons are quite fair.

We have revised this statement and elaborated on the selection of this District; namely, it’s representation of the larger country. 

Line 135 – I wonder why individuals living within a 1 km radius of health facilities were only eligible participants as the distance to health facilities could be a barrier, especially in hilly regions of Nepal.

We have noted in the paper that the catchment areas of the health facilities are 1-km in radius. Therefore, sampling outside of this radius would have resulted in including participants who did not visit the study facilities. 

Line 138 - Are there any reasons why mothers were not included in in-depth interviews?

In the Objectives section, we have clarified that this formative work is informing a social norms-based intervention focusing on mothers of young children and health workers. Therefore, a key domain of interest was norms related to immunization, which we convey through the descriptions of the instruments. We have clarified in the Participants section that for mothers, we were particularly interested in understanding the ways in which they communicate with each other about immunization. 

Line 172 – Data collection procedure?? Also, please mention the data collection period as it could be valuable data.

The first part of the comment was not entirely clear to us (but we did add that data were collected “in-person.” And we have added the data collection period, as suggested. 

Lines 190 to 192 – Could you please associate the reference cited with this study? If not, you need to mention the procedure here clearly.

This reference was cited earlier in the paper on page 6 under the Study Objectives section as the protocol for the Rejoice Architecture Project, the parent study of this research. 

Lines 195 to 197– Providing enough context and clarity doesn't justify why member checking was not done to increase the credibility of the research.

This is a limitation of the study, and we have noted it in the Limitation section

Line 199 – Does it mean each interview and focus group discussion covered a separate topic? If so, it limits the transferability of the research. Can you please explain?

No. In the Instruments section, we describe that all instruments cover the same three topics (i.e., facilitators, barriers, and social norms), and that each instrument also covers other topics in addition to these three topics (e.g., the in-depth interview guide covers family experiences, the key informant guide covers local coverage, etc.). 

Line 200 – The instruments should only be a guideline. It is not clear whether there was a separate guideline for each topic. Also, please include interview guidelines as an additional file.

Each topic had its own semi-structured questions with suggested probes. Data collectors were trained to follow the instruments as a guide, but let the conversation flow as needed, as we indicate on page 11 of the Instruments section. 

The direct link to the registered interview and focus group instruments have been added to the references (ref no. 24). We have also uploaded the instruments as supplemental files.

Results

Line 307 - Here, the gender norm of women being the primary caretaker does not justify being a facilitating factor to increase vaccine coverage. Instead, the link between women receiving immunisation awareness when participating in mothers' groups and being the child's primary caretaker increased the likelihood of vaccinating their child.

This section was aiming to describe how gender norms influenced the immunization practices and experiences in the community, rather than the likelihood of getting vaccinated. We have removed it, as it was not a strong theme in regard to facilitators of immunization. 

Line 344 – The last paragraph within this sub-theme is more of accountability issue and mistrust as represented by the theme, not a power imbalance.

We re-evaluated this paragraph and have moved it to the “Clients perceive micro-aggressions” sub-theme, as it depicts clients adapting their behavior (i.e., going to private facilities) to avoid feelings of discrimination and disrespect at public facilities. 

Line 364 – Is 5-hours a regular office opening hours, or was it because of the absence of health workers in the facilities to provide service. Immunisation has been covered mainly as an outreach service in communities in Nepal. Does the above-mentioned service hour refer to it? Would you please address this clearly?

Health facilities offer many other services than immunization, and so their opening hours are not a reflection of immunization outreach. Health posts should be open for at least 6 hours per day from 10am-4pm. Although we cannot confirm the exact hours these facilities were open, we have edited this sentence to clarify that, in general, participants were unsatisfied with the few hours per day the health facilities were open for services. 

Line 380 to 383 – It is not acceptable to include information from the researcher's perspective in the result section, especially in this type of study. If it is explained based on the participants' response, please remove "Generally speaking" and support this claim with participants' quote.

This paragraph is not from the researcher’s perspective, but a summary of the general findings from health worker reports. We have revised this paragraph to make this clear.

Line 384 to 389 – This is going out of context unless linked with the study's objective. Are they talking about addressing adverse effects of immunisation or the mistrust due to previous experiences on immunising their child? Or are they just expressing their other concerns related to health care services?

We have removed this paragraph as it did not align well with the sub-heading and was not a strong enough or relevant enough finding to pull out on its own. 

Line 394 – This sentence does not reflect the essence of the quote here. It is more about not getting other services if not fully vaccinate their child. Refusing to immunise a child by health worker is a sensitive claim, and therefore a quote is required.

We have revised this sentence to more accurately set up the following quotes.

Line 431 – Why family planning devices is affecting immunisation efforts?

Family planning devices are just one example of how a lack of resources is negatively affecting service delivery. To avoid confusion, we have removed this example. 

Line 445 to 453 – Why the results are not linked with immunisation here? It looks more like an interview related to the condition of the health care facility rather than how the health facility environment affects vaccine uptake.

The health facility environment affects not only immunization service delivery, but delivery of all services offered in those facilities. We have added a clause to make this salient. 

Line 454 – Provider or Provider’s

Discussion

Line 518 – Time hasn't been explained clearly as a barrier in the result section in the way it is mentioned here in the discussion.

This section has been removed. 

Line 524 to 526 – It is expressed from the key informants on the supply side rather than the family caregiver in the result section. It is misleading.

This section has been removed.

Overall, the discussion should be coherently organised as per the objective of the study and rewritten.

Please see our response above. 

Limitations

Line 588 to 589 – "It is also a fertile district able to grow and export cash crops to neighbouring cities of 589 Kathmandu and Hetauda" – This sentence has nothing to do with this study unless the socio-economic status of study participants is linked with vaccine uptake.

This sentence has been removed.

There could be more limitation

– one could be member checking (as mentioned earlier)

This has been noted

- Including participants within 1 km radius excluded participants who could be affected by travel time to health facilities

This has been noted

- No information is provided about the participants' backgrounds. It would be worthy of increasing the transferability of the study findings.

This is a tough one. We believe the most relevant background is that Makwanpur is a district with much greater access to urban facilities (in Kathmandu and Hetauda) than other districts in Nepal. This reduces generalizability, but the title now contains the name of the district, which communicates where the study was done (and hence the extent of appropriate generalizability). 

- Was the number of participants finalised beforehand? Or was it based on data saturation?

This was done beforehand, because our IRB requires us to provide information about the sample size.

Conclusion

Line 594 – "stocking adequate doses of vaccines" - not mentioned in the result section and, therefore, cannot be included in the conclusion

Line 595 – "people's perceptions that many others do not fully vaccinate their children" – How is this a barrier? And why is it on conclusion rather than other important barriers discussed in the result section?

Line 596 – "feelings of experienced disrespect" – What does this mean?

We have revised our examples here to reflect the key themes from the results section. 

Line 608 – There's nowhere in the results section that travel time is a barrier.

We have removed this barrier, as we do not delve into this issue in this paper.

Line 617 – Who are these marginalised groups? There is no identification of marginalised groups' deviant behaviours in the results.

We have revised this to reference clients more generally. 

Overall, the conclusion should be rewritten.

Response to Reviewer 2’s Comments:

The sample size is somewhat an issue, given the information that the work is trying to capture.

We assume that the reviewer is noting that the sample size was not large enough. This is difficult to ascertain, of course, but we did find repetitions of themes in the data, suggesting that we were close to saturation. We should also note that the title of the paper now has the name of the district, which communicates that findings pertain to the one district.

The one district (out of 77 in the country) that was chosen is in close proximity to the capital of Nepal, Kathmandu and as such it may not truly represent the situation prevalent in rest of Nepal. The authors should make sure that this is identified as a major limitation of the study.

We have elaborated on the representativeness of Makwanapur in the Study Setting section. 

The current title of the study and its objectives do not match the results provided. Perhaps the title and objectives of the study should be revised to truly reflect the work carried out.

The title and objectives are addressed above.

---

## [Decision Letter · Decision Letter 1]

2 Dec 2021

PONE-D-21-16867R1The last stretch: Barriers to and facilitators of full immunization among children in Nepal’s Makwanpur District, results from a qualitative studyPLOS ONE

Dear Dr. Paul,

Thank you for submitting your manuscript to PLOS ONE. After careful consideration, we feel that it has merit but does not fully meet PLOS ONE’s publication criteria as it currently stands. Therefore, we invite you to submit a slightly revised version of the manuscript that addresses the points raised during the review process by Reviewer #1 (only one review was sought).  I, as the Academic Editor, concur with Reviewer #1.  I also carefully reviewed your revision that, I believe, was responsive to the reviews of your original submission.  Now only a few suggested modifications are pending with you prior to revision and resubmitting.

We look forward to receiving your revised manuscript.

Kind regards,

Andy Stergachis, PhD

Academic Editor

PLOS ONE

Journal Requirements:

Reviewers' comments:

Reviewer's Responses to Questions

**Comments to the Author**

1. If the authors have adequately addressed your comments raised in a previous round of review and you feel that this manuscript is now acceptable for publication, you may indicate that here to bypass the “Comments to the Author” section, enter your conflict of interest statement in the “Confidential to Editor” section, and submit your "Accept" recommendation.

Reviewer #1: All comments have been addressed

2. Is the manuscript technically sound, and do the data support the conclusions?

Reviewer #1: Yes

3. Has the statistical analysis been performed appropriately and rigorously? 

Reviewer #1: N/A

4. Have the authors made all data underlying the findings in their manuscript fully available?

Reviewer #1: Yes

5. Is the manuscript presented in an intelligible fashion and written in standard English?

Reviewer #1: Yes

6. Review Comments to the Author

Reviewer #1: I highly appreciate the authors’ for thoroughly revisiting the manuscript as required. I have very few minor disagreements.

1. Line 84

I still disagree that this sentence—Nepal requires seven separate visits over 15 months to complete the full schedule [12,13]—is a generalised sentence that represents results from other low- and middle-income countries as it specifically mentions about Nepal. Therefore, I would suggest to either remove the citation from Kenya or make the sentence more generalised.

2. Line 178

The title of this section is only ‘Procedure’. As it is the data collection procedure, it would be better to have the clear title for this section.

3. The same information is repeated in line 254 (generational shift in attitudes toward immunization) and line 256. Please avoid repeating the same information, especially within the same section.

7. PLOS authors have the option to publish the peer review history of their article (what does this mean?). If published, this will include your full peer review and any attached files.

Reviewer #1: No

---

## [Author Response · Author response to Decision Letter 1]

10 Dec 2021

Journal Requirements: Please review your reference list to ensure that it is complete and correct. If you have cited papers that have been retracted, please include the rationale for doing so in the manuscript text, or remove these references and replace them with relevant current references. Any changes to the reference list should be mentioned in the rebuttal letter that accompanies your revised manuscript. If you need to cite a retracted article, indicate the article’s retracted status in the References list and also include a citation and full reference for the retraction notice.

In this revision, we have made no changes to our reference list. No references have been retracted. We confirm that it is complete and correct. 

Comments to the Author

6. Review Comments to the Author

Reviewer #1: I highly appreciate the authors’ for thoroughly revisiting the manuscript as required. I have very few minor disagreements.

1. Line 84

I still disagree that this sentence—Nepal requires seven separate visits over 15 months to complete the full schedule [12,13]—is a generalised sentence that represents results from other low- and middle-income countries as it specifically mentions about Nepal. Therefore, I would suggest to either remove the citation from Kenya or make the sentence more generalised.

Thank you for pointing this out. We agree. We have removed this reference from line 84. 

2. Line 178

The title of this section is only ‘Procedure’. As it is the data collection procedure, it would be better to have the clear title for this section.

Thank you for clarifying this comment. We have updated the title of this section to be “Data Collection Procedure”. 

3. The same information is repeated in line 254 (generational shift in attitudes toward immunization) and line 256. Please avoid repeating the same information, especially within the same section.

Thank you for pointing this out. We have removed the sentence which began on line 256.

---

## [Editor Report · Decision Letter 2]

14 Dec 2021

The last stretch: Barriers to and facilitators of full immunization among children in Nepal’s Makwanpur District, results from a qualitative study

PONE-D-21-16867R2

Dear Dr. Paul,

We’re pleased to inform you that your manuscript has been judged scientifically suitable for publication and will be formally accepted for publication once it meets all outstanding technical requirements.

Kind regards,

Andy Stergachis, PhD

Guest Editor

PLOS ONE

Additional Editor Comments (optional):

This is a improved revision, fully responsive to suggested edits proposed by the reviewer with concurrence from the Guest Editor.
---

## [Editor Report · Acceptance letter]

15 Dec 2021

PONE-D-21-16867R2 

The last stretch: Barriers to and facilitators of full immunization among children in Nepal’s Makwanpur District, results from a qualitative study 

Dear Dr. Paul:

I'm pleased to inform you that your manuscript has been deemed suitable for publication in PLOS ONE. Congratulations! Your manuscript is now with our production department. 

Kind regards, 

on behalf of

Dr. Andy Stergachis 

Guest Editor

PLOS ONE